# Fully Atomistic Molecular Dynamics Simulation of a TIPS-Pentacene:Polystyrene Mixed Film Obtained via the Solution Process

**DOI:** 10.3390/nano13020312

**Published:** 2023-01-11

**Authors:** Tomoka Suzuki, Antonio De Nicola, Tomoharu Okada, Hiroyuki Matsui

**Affiliations:** 1Research Center for Organic Electronics (ROEL), Yamagata University, Jonan 4-3-16, Yonezawa 992-8510, Japan; 2Scuola Superiore Meridionale, Largo San Marcellino 10, 80138 Napoli, Italy

**Keywords:** organic semiconductors, molecular dynamics simulation, printed electronics, thin films

## Abstract

Organic thin-film transistors using small-molecule semiconductor materials such as 6,13-bis(triisopropylsilylethynyl)pentacene (TIPS-P) have been recently studied for the production of flexible and printed electronic devices. Blending a semiconductor with an insulating polymer, such as polystyrene, is known to improve the device performance; however, its molecular-level structure remains unknown. In this study, we performed molecular dynamics (MD) simulations on a mixed system of TIPS-P and atactic polystyrene (aPS) with fully atomistic models to understand the structure of the mixed thin film at the molecular level and the influence on the device properties. To reproduce the deposition from the solution, we gradually reduced the number of toluene molecules in the simulation. The dynamic characteristics of the system, mean squared displacement, diffusion coefficient, density profile, and P_2_ order parameter were analyzed. Some of the simulated systems reached the equilibrium state. In these systems, the simulated structures suggested the presence of more TIPS-P molecules on the surface than inside the bulk, even at the low molecular weight of aPS, where phase separation was not observed experimentally. The results of the fully atomistic MD simulations are also a basis for the coarse-grained model to increase the speed of the MD simulation.

## 1. Introduction

Flexible and printed organic electronic devices have been extensively studied over the last three decades [1] because of their unique mechanical features, such as their ability to bend, stick, and be disposed of [2,3,4], which are different from those of conventional electronic devices based on rigid inorganic semiconductors. These unique features make flexible electronic devices suitable for healthcare applications such as biosensing [5,6] and electronic organs [7,8,9] (bioelectronics). The two main trends in the research field of flexible electronic devices are (i) reducing the thickness of films integrating or constituting the electronic components [10] and (ii) developing new materials suitable for producing flexible electronic devices [11,12,13]. One of the molecules used as organic semiconductors in flexible electronic devices is pentacene, which exhibits a high carrier mobility [14,15] (~1.2 cm^2^/Vs) required for applications in electronic devices. However, the material requires a vacuum deposition process, which notably increases the cost [16,17]. An alternative to pentacene is 6,13-bis(triisopropylsilylethynyl)pentacene (TIPS-P), which has been developed as a highly soluble semiconductor for flexible electronic devices [18,19]. In particular, TIPS-P is used as the active layer of organic thin-film transistors as a small-molecule semiconductor owing to its solubility and conductivity [17,20]. Owing to its high solubility, TIPS-P can be deposited via solution processes at room temperature at a lower cost than pentacene and is suitable for large-area electronic products [16,18,21]. TIPS-P also has a high carrier mobility, exceeding 1 cm^2^/Vs [17]. Ohe et al. experimentally reported that blending TIPS-P with an insulating polymer, poly-α-methylstyrene (PαMS), can improve the performance and homogeneity of organic thin-film transistors [17]. In their experiments, spin-coating a solution of TIPS-P and PαMS in toluene formed a tri-layer structure of a TIPS-P layer/mixed layer of TIPS-P and PαMS/TIPS-P layer via spontaneous phase separation if the weight-average molecular weight (M_w_) of PαMS was larger than 28,200 [17]. The blend of TIPS-P and PαMS with a low M_w_ of 2200 resulted in lower device performance without phase separation. However, the reason why the insulating polymer affects the device performance, as well as its interfacial structures, has not been revealed to date. The technique of blending small-molecule semiconductors with insulating polymers has been extended to different combinations of materials [22].

In this study, we performed fully atomistic molecular dynamics (MD) simulations of a TIPS-P:atactic polystyrene mixed film to understand its structure and interactions at the molecular level. The thin-film formation from the solution was mimicked by gradually reducing the toluene content in the system. Using MD simulations with atomistic models makes it possible to reproduce the molecular state at the surface in more detail, which is otherwise difficult to accomplish by using experimental approaches such as atomic force microscopy and X-ray diffraction. To date, trial-and-error experiments based on device features have been conducted for the selection of polymer materials. Elucidating the mechanism from the results of MD simulations will guide material selection. The results of the fully atomistic MD simulation are also a basis for the coarse-grained (CG) model to increase the speed of the MD simulation.

## 2. Methods

### 2.1. Atomistic Models

In this study, atomistic models of atactic polystyrene (aPS, C_8*n*_H_8*n*+2_) [23], TIPS-P (C_44_H_54_Si_2_) [24], and toluene (C_7_H_8_) [25] were used. Figure 1 shows the molecular formulas of aPS, TIPS-P, and toluene. For all the considered models, OPLS-AA force field parameters were used. A short oligomer of aPS (M_w_ = 1056, C_80_H_82_), composed of ten repeating units, was selected because of the limitation of the calculation cost and to have access to the characteristic relaxation time of aPS occurring at different timescales, which can be achieved by full-atomistic simulations. The aPS model, which is able to correctly reproduce the structural and dynamic properties of the polymer bulk, was taken from the literature [23,26]. The TIPS-P model, proposed by Steiner et al., was used to study the crystalline structure of TIPS-P [24].

### 2.2. Calculation for the Pristine TIPS-P Crystal

To validate the force field for TIPS-P, an MD simulation of pristine TIPS-P crystals without PS and toluene was performed. The 10 × 8 × 4 supercell of the experimental structure [27] was used as the initial configuration. Energy minimization was performed using 5000 steps of the steepest descent algorithm. A time step of 2 fs for this system and a cut-off distance for the Lennard-Jones non-bonded interaction of 1.2 nm were set. Periodic boundary conditions were considered in all directions (*x*, *y*, and *z*). For the production run, the temperature was kept constant using a velocity-rescaling algorithm [28] with a characteristic relaxation time *τ_T_* = 0.02 ps and a fixed volume. Simulated annealing was performed from 0 to 200 K every 100 ps in 50 K increments, with a simulation time of 1 ns. Chemical structures of all molecules used in the MD simulations are reported in Figure 1.

### 2.3. Calculation for the Mixed Solution System

The free software GROMACS [29] (Ver. 2016.4) was used for all simulations, and the temperature was kept constant using a velocity-rescaling algorithm [28] with a characteristic relaxation time *τ_T_* = 0.02 ps. The pressure was kept constant using the Berendsen weak-coupling scheme, and the box was scaled isotropically with a characteristic relaxation time *τ_P_* = 0.3 ps. Periodic boundary conditions were used in all directions (*x*, *y*, and *z*). A time step of 2 fs was set for all systems. The cut-off distance for the Lennard-Jones non-bonded interactions was set to 1.2 nm. The electrostatic interactions were computed using the Ewald particle mesh (mesh spacing in Fourier space: 0.12 nm) [30,31]. All the bonds involving hydrogen atoms were constrained using the LINCS algorithm [32]. For all the initial coordinate sets, the energy of the systems was minimized by performing 5000 steps of the steepest descent algorithm. On each initial set of coordinates, an NPT simulation of the bulk system was performed until the total mass density reached equilibrium. As a further validation model test, we performed an additional run in the NPT ensemble by using a different barostat algorithm, the Parrinello–Rahman algorithm (with same target temperature and pressure) [33]. For the additional run, we used the last well-equilibrated configuration obtained from the set-up with a Berendsen barostat. In Figure 2, the comparison of the time evolution of the mass density of the system calculated for the two barostat algorithms is reported. As can be seen from the trends in Figure 2, we can assume that the key interactions reproduced by the Berendsen set-up are in reasonable agreement with the Parrinello–Rahman set-up (which is known to be accurate in the reproduction of the canonical ensemble) [34,35]. For both algorithms, the mass density converges to the same equilibrium value.

After the NPT run, we performed an NVT simulation by taking the last configuration of the NPT simulation and extending the box side length up to 20 nm to create the interface under vacuum. The toluene content was gradually reduced from 50 to 30, 20, 10, and 5 *w*/*w*%. A constant number of TIPS-P molecules and aPS chains was used (Table 1). First, we prepared a 50 *w*/*w*% toluene system and randomly set all molecules in the simulation box without a vacuum phase. Minimization and NPT simulations were performed for the bulk, followed by an NVT simulation, in which the interface of the bulk with vacuum was included. Considering the experimental conditions [17], the simulations were performed at room temperature (300 K) and the annealing temperature (333 K). A trajectory of 1 µs was accumulated for all systems. Starting from a set of coordinates (taken at equilibrium) of the system at a higher toluene content, some toluene molecules were removed from the system according to the new (lower) concentration. The same protocol, minimization, and NPT run were then applied, and a long NVT production run (1 µs with the bulk/vacuum interface) was performed. This procedure was performed until a system with a toluene content of 5 *w*/*w*% was simulated. The compositions and total times of the simulated systems are listed in Table 1.

## 3. Results and Discussions

### 3.1. Pristine TIPS-P Crystal for Validation of the Force Field

Different planar views of a representative equilibrium configuration of the simulated TIPS-P crystal (Figure 3a) are compared with the crystalline structure of the pristine TIPS-P in the literature [27] (Figure 3b). In panel 3c of the same figure, the time evolution of the powder X-ray diffraction pattern calculated from the MD simulation is compared with the experimental one (black segmented line). As the simulation time increases, peak positions shift to lower 2*θ* values (see the sequence of calculated patterns from 3 to 50 ns), indicating an increase in the lattice constant parameters (from ~1 to ~10%). The average values of the lattice constants obtained at equilibrium are: *a* = 7.70 Å, *b* = 7.83 Å, *c* = 16.82 Å, *α* = 93.3°, *β* = 89.5°, and *γ* = 91.1°, which reasonably agree with the experimental values: *a* = 7.55 Å, *b* = 7.73 Å, *c* = 16.76 Å, *α* = 89.5°, *β* = 78.7°, and *γ* = 84.0° [27]. These results indicate the validity of the adopted force field to simulate the crystalline structure of TIPS-P.

### 3.2. TIPS-P:aPS:Toluene Ternary System

#### 3.2.1. Mean Squared Displacement

The mean squared displacement (MSD) [37,38,39] can provide an approximate estimation of the capability of sampling the entire phase space of the system for each component. Figure 4 shows the time-dependence of the MSDs in the direction normal to the bulk/vacuum interface. At toluene content of 50 *w*/*w*%, the MSDs of TIPS-P and aPS saturated at 15 and 10 nm^2^, respectively. The thickness *T* of the ternary system was approximately 10 nm at all times. If each molecule has enough time to diffuse within the thickness, the MSD is expected to be
$$MSD=\int_{0}^{T}\int_{0}^{T}\frac{{\left(s-t\right)}^{2}}{T^{2}}dsdt=\frac{T^{2}}{6}$$
where *s* and *t* are the initial and final positions of a molecule, and 1/*T* gives the uniform probability density of the position between 0 and *T*. This formula and *T* = 10 nm give MSD = 16.7 nm^2^. Therefore, the saturation of MSDs at 15 and 10 nm^2^ indicates that the TIPS-P and aPS had enough time to diffuse within the thickness in the systems with toluene content of 50 *w*/*w*%. However, the MSDs in the systems with toluene content of 30 *w*/*w*% or less did not saturate within the simulated time, which indicates that systems with lower toluene content require simulation times longer than 1000 ns. The diffusion coefficient of each component was also calculated using the least-squares method of linear fitting to the time-MSD curve. The diffusion constants for all toluene contents are listed in Table 2.

#### 3.2.2. Polymer Configuration

To evaluate the effectiveness of sampling the polymer configurations, we computed the end-to-end relaxation time for the aPS chains. The autocorrelation function $C(t)=\frac{\left(R(\tau )R(\tau +t)\right)}{\left(R^{2}\right)}$ was used to compute the relation time (Figure 5), where *R* is the length of the end-to-end distance vector, and the brackets indicate the time average. Then, the relaxation time $\left(\tau_{PS}\right)$ was computed by integrating the stretched exponential function fitted to the autocorrelation functions, as described in Equation (1).


(1)
$$\tau_{PS}=\int_{0}^{\infty }\exp\left[-{\left(\frac{t}{\alpha }\right)}^{\beta }\right]dt=\frac{\alpha }{\beta }Г\left(\frac{1}{\beta }\right)$$


The $\tau_{PS}$ values are listed in Table 3. As the toluene content decreased from 50 to 5 *w*/*w*%, the relaxation time of aPS increased from ~0.6 to 200 ns. In addition, shorter relaxation times were observed at higher temperatures. The total simulation time was approximately 4–5 times the relaxation time $\tau_{PS}$ at 5 *w*/*w*% toluene. From these data, we can conclude that the sampling of chain conformations is sufficient for all the toluene contents.

#### 3.2.3. Density Profile

From the results of MSD in Figure 4, the system with 50 *w*/*w*% toluene was considered under the equilibrium after 500 ns. The density profile of each component was calculated in 50 ns increments from 0 to 1000 ns. Figure 6 shows the density profiles, computed along the normal direction of the bulk/vacuum interface, of the system with 50 *w*/*w*% toluene at 50, 500, and 1000 ns. Figure 7 shows the configurations of the same system. The density profile indicates that the concentration of TIPS-P is higher within 1 nm of the surface. However, experimental studies have reported no phase separation when the M_w_ of PαMS is 2000 or lower [17]. This is because the experimental method does not have a high spatial resolution of 1 nm. Our simulation reveals, for the first time, that the surface contains more TIPS-P molecules than the film, even at a low M_w_ of 1056. Such phenomena can be considered a precursor of phase separation into a tri-layer structure at the high molecular weight of aPS.

#### 3.2.4. P_2_ Order Parameter

To investigate the orientation in different regions in the bulk and at the bulk/vacuum interface, we calculated the order parameter P_2_ [40]. The order parameter P_2_ is defined as $P_{2}=\frac{3〈\cos^{2}\;\theta 〉-1}{2}$, where *θ* is the angle between the molecular vector of TIPS-P and the direction normal to the bulk/vacuum interface. P_2_ = −0.5 if the molecular vector is perpendicular to the interface, and P_2_ = 1 if the molecular vector is parallel to the interface. If the molecules are randomly oriented, P_2_ = 0. The TIPS-P molecular vector was defined using the positions of the two silicon atoms.

The order parameter P_2_ was calculated at each slice (thickness: 2 nm) parallel to the surface for a system with 50 *w*/*w*% toluene at 300 K at 50, 500, and 1000 ns (see Figure 6). As can be seen from the profiles, the P_2_ values were almost equal to 0 under all these conditions, which indicates the random orientation of the investigated molecules. This is consistent with the experimental result for PαMS with M_w_ ≤ 2000 [17].

#### 3.2.5. Surface Profile

Some indication of the roughness of the surface can be obtained from the profile decay. The roughness of the surface can be measured using the root mean squared roughness R_q_ [41], where R_q_ is defined as (2)$$R_{q}=\sqrt{\frac{1}{N^{2}}\sum_{i(x)=0}^{N}\sum_{j(y)=0}^{N}{\left(h_{ij}-\bar{h}\right)}^{2}}$$


To this end, the system was divided into 10 × 10 bins in the direction parallel to the bulk/vacuum interface (*x*, *y*). For each bin, the largest height *h*(x, y) was calculated by considering the atom coordinates of both TIPS-P and aPS. The term $\bar{h}$ in Equation (2) is the average height of the surface, computed by considering all 10 × 10 bins. The R_q_ values computed for all systems are compared in Figure 8.

The main result emerging from the roughness calculation confirms that the TIPS-P and aPS molecules were arranged in a manner that minimizes R_q_. R_q_ was the highest with 50 *w*/*w*% toluene (0.31 nm). R_q_ decreased as the toluene content decreased, and finally, R_q_ was 0.17 nm with 5 *w*/*w*% toluene. The range of R_q_ agreed with the experimental range (0.2–0.5 nm) [42]. The increase in the amount of toluene perturbed the roughness of both molecules. The temperature did not seem to affect the roughness significantly.

## 4. Conclusions

We performed a fully atomistic MD simulation for a blend system of TIPS-P and aPS in toluene. The evaporation of toluene was mimicked by a gradual reduction in the number of toluene molecules. The MSDs and diffusion coefficients for all systems indicated that equilibrium states were reached under two conditions: 50 *w*/*w*% at 300 K and 50 *w*/*w*% at 333 K. The density profile revealed that the concentration of TIPS-P was higher at the interface with vacuum, whereas no clear phase separation was observed because of the low molecular weight of aPS. The high concentration of TIPS-P at the surface is advantageous to forming conductive paths in organic thin-film transistors. The random orientation of the TIPS-P molecules was observed, which is consistent with the experimental results. The analysis of the surface roughness indicated that the TIPS-P molecules were arranged in a manner that produced a smooth surface, which is compatible with the high ordering of the molecules. Further studies with a higher M_w_ would clarify the film structure with improved mobility in the experiments. However, this requires a CG model instead of a fully atomistic model to reduce the calculation cost and to gain access to the intrinsically faster dynamics of the CG models. The present results can be used to develop CG models and to approach the problem with a multiscale strategy, reintroducing atomic details via backmapping methodology where needed.

## Figures and Tables

**Figure 1 nanomaterials-13-00312-f001:**
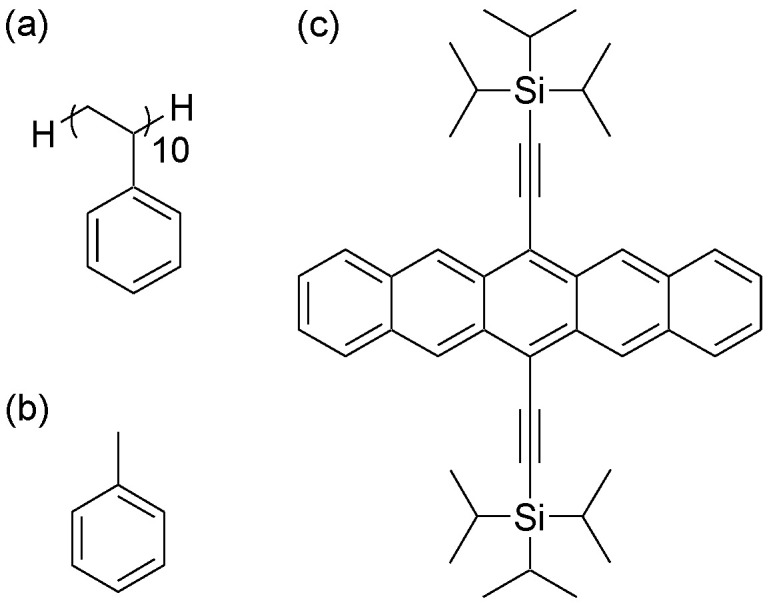
Molecular formulas of (**a**) polystyrene, C_80_H_82_; (**b**) toluene, C_7_H_8_; and (**c**) TIPS-P, C_44_H_54_Si_2_.

**Figure 2 nanomaterials-13-00312-f002:**
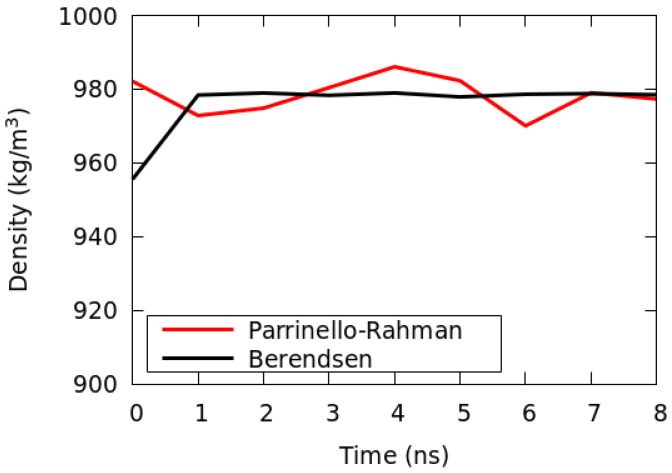
Time behavior of mass density for the Berendsen (black) and Parrinello–Rahman (red) barostat algorithms. Only the last 8 ns of trajectory, starting from a well-equilibrated sample, are shown.

**Figure 3 nanomaterials-13-00312-f003:**
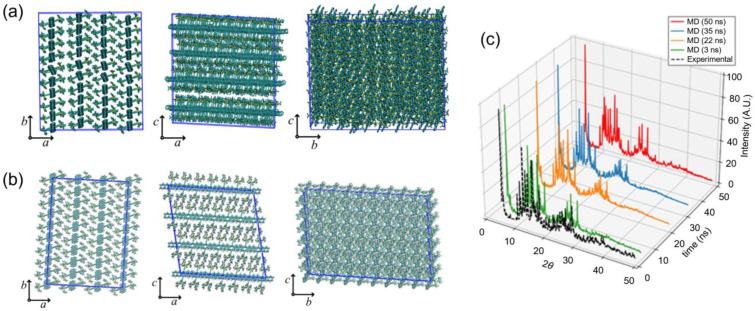
(**a**) The different views of a representative equilibrium configuration of TIPS-P crystalline structure obtained via MD simulation. (**b**) The different views of the TIPS-P crystalline structure obtained via experiment [36]. (**c**) Comparison between simulated and experimental [36] powder X-ray diffraction patterns of the TIPS-P crystal.

**Figure 4 nanomaterials-13-00312-f004:**
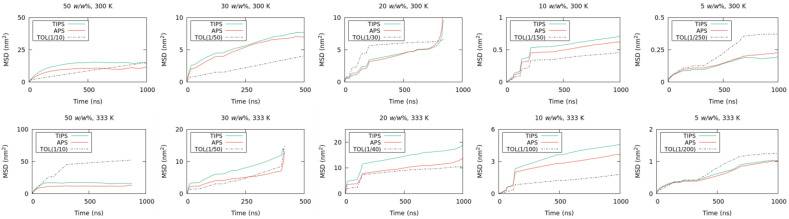
Time-dependence of the MSDs at different toluene contents and temperatures. TIPS-P (green), aPS (red), and toluene (black, dotted line).

**Figure 5 nanomaterials-13-00312-f005:**
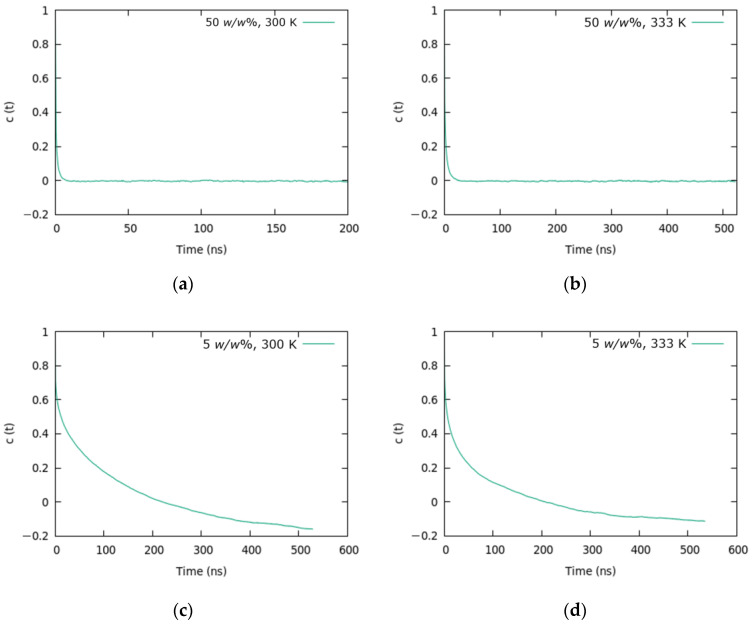
Autocorrelation functions of the end-to-end distance for the systems of 50 *w*/*w*% toluene at 300 K and 333 K (**a**,**b**) and 5 *w*/*w*% toluene at 300 K and 333 K (**c**,**d**).

**Figure 6 nanomaterials-13-00312-f006:**
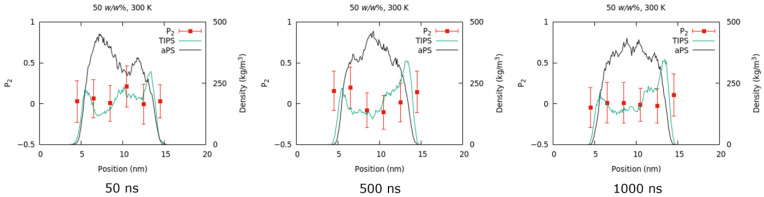
Density profiles of aPS (black curve) and TIPS-P (green curve) together with the P_2_ order parameters of TIPS-P for the systems with 50 *w*/*w*% toluene at 300 K.

**Figure 7 nanomaterials-13-00312-f007:**
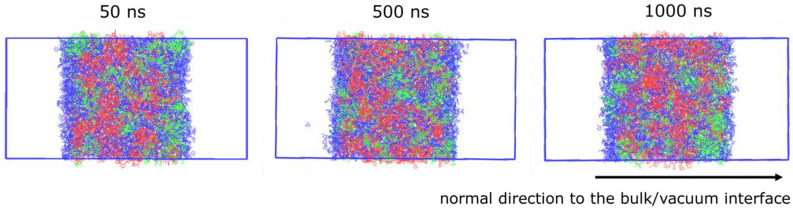
Configurations of the system containing 50 *w*/*w*% toluene at 300 K at 50, 500, and 1000 ns. Owing to the applied periodic boundary condition, there are two bulk/vacuum interfaces in our simulation box. (Red: aPS, blue: toluene, and green: TIPS-P).

**Figure 8 nanomaterials-13-00312-f008:**
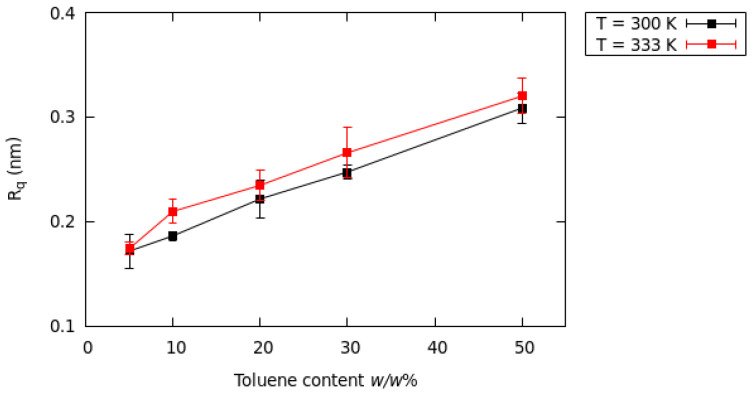
Surface roughness R_q_ (nm) of the mixed film with standard error (sample size is 3) at different toluene contents.

**Table 1 nanomaterials-13-00312-t001:** Composition of the simulated systems.

*w*/*w*% Toluene	# of Toluene	# of TIPS-P	# of aPS Chain	Box [*x*, *y*, *z*] (nm)	Total Time (µs)
50	2700	150	150	9.5; 19.0; 9.5	1
30	1620	150	150	8.7; 8.7; 40.0	1
20	1080	150	150	7.8; 7.8; 30.0	1
10	540	150	150	7.9; 7.9; 20.0	1
5	270	150	150	7.6; 7.6; 20.0	1

**Table 2 nanomaterials-13-00312-t002:** Diffusion coefficients with standard error (sample size is 4) inside the bulk in the normal direction of the surface for toluene, TIPS-P, and aPS.

Toluene Content (*w*/*w*%)	Toluene (cm^2^/s)		TIPS-P (cm^2^/s)		aPS (cm^2^/s)	
	300 K	333 K	300 K	333 K	300 K	333 K
50	(5.63 ± 0.07) × 10^−6^	(1.33 ± 0.01) × 10^−5^	(8.0 ± 0.7) × 10^−7^	(1.29 ± 0.04) × 10^−6^	(5.9 ± 0.2) × 10^−7^	(10 ± 1) × 10^−7^
20	(2.5 ± 0.5) × 10^−6^	(3 ± 2) × 10^−6^	(7.2 ± 0.5) × 10^−8^	(2.1 ± 0.4) × 10^−7^	(5.1 ± 0.4) × 10_−8_	(2.0 ± 0.3) × 10^−7^
10	(1.6 ± 0.1) × 10^−6^	(3.4 ± 0.8) × 10^−6^	(1.8 ± 0.3) × 10^−8^	(7 ± 1) × 10^−8^	(1.17 ± 0.08) × 10^−8^	(6.8 ± 0.8) × 10^−8^
5	(9 ± 3) × 10^−7^	(1.8 ± 0.8) × 10^−6^	(4 ± 1) × 10^−9^	(1.5 ± 0.4) × 10^−8^	(4 ± 1) × 10^−9^	(1.2 ± 0.3) × 10^−8^

**Table 3 nanomaterials-13-00312-t003:** Relaxation time ($\tau_{PS}$) computed for systems at 50 and 5 *w*/*w*% toluene. Standard errors were calculated by dividing the 150 aPS chains into 10 groups.

Toluene Content (*w*/*w*%)	Temperature (K)	$\tau_{PS}$ (ns)
50	300	0.64 ± 0.03
50	333	2.2 ± 0.1
5	300	240 ± 20
5	333	190 ± 30

## Data Availability

Not applicable.

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
