# Peer review of "Fully Atomistic Molecular Dynamics Simulation of a TIPS-Pentacene:Polystyrene Mixed Film Obtained via the Solution Process"

_nanomaterials, 2023, doi:10.3390/nano13020312_

Round 1

Reviewer 1 Report

This article is devoted to the simulation of atomic molecular dynamics of a mixed film TIPS-pentacene: polystyrene. The article is new and interesting. The relevance is beyond doubt. There are some important points that it is desirable to improve:

1. The main formulas of the calculated and indicated substances must be indicated in the text of the article.

2. It is desirable to expand the abstract.

3. Did the authors use calculations for several solvents?

4. Theoretical calculations are made at a good level. However, I strongly recommend adding more comparisons with experimental data from the literature.

5. In addition, more alignment of the calculations obtained by the authors with data from the literature is needed.

6. Conclusions can be made more concise.

7. Please cite: 10.1016/j.molliq.2022.119859.

Author Response

Point 1: The main formulas of the calculated and indicated substances must be indicated in the text of the article.

Response 1: We thank the reviewer for the important suggestion. We modified Figure 1 so as to make atomic elements and bond types clear. Also, the chemical formulas are added in the body text and figure caption.

Point 2: It is desirable to expand the abstract.

Response 2: We thank the reviewer for the helpful comment. Because of the length limitation (200 words), we could not expand the abstract.

Point 3: Did the authors use calculations for several solvents?

Response 3: We used only toluene as a solvent. No other solvent has been employed. Toluene was chosen because it is a typical aromatic solvent in printed electronics, dissolves aromatic organic semiconductors well, and was used in the reference experimental work [17]. Benzene has a simpler structure but is more toxic and is not popular in printed electronics.

Point 4: Theoretical calculations are made at a good level. However, I strongly recommend adding more comparisons with experimental data from the literature.

Response 4: We thank the reviewer for the important comment. We made three comparisons between the experimental and simulated results. The first is the crystal structure of TIPS-P for the validation of the force field (Figure 2). The simulated crystal structure, including XRD pattern and lattice constants, had a good agreement with the experiment. The second is the density profile of the TIPS-P in Figure 5. In contrast to the homogeneous distribution of TIPS-P for the small molecular weight (Mw = 1,000) in experiment, we could observe higher density of TIPS-P at the surface because of the higher spatial resolution. The third is the molecular orientation. The experiment with small-Mw polymer resulted in unoriented and amorphous film, which is consistent with our simulation.

Point 5: In addition, more alignment of the calculations obtained by the authors with data from the literature is needed.

Response 5: We thank the reviewer for the helpful comment. Although we could not compare our simulated results with the literature, we compared our results with an analytical model to better understand the behavior of the time-dependence of the MSD. We hope that the new analysis has improved the validity of our results.

Point 6: Conclusions can be made more concise.

Response 6: We thank the reviewer for the helpful suggestion. We made the conclusion paragraph more concise.

Point 7: Please cite: 10.1016/j.molliq.2022.119859.

Response 7: Somehow we could not find a relevance of this article to our manuscript. We would appreciate if the reviewer can specify the position to cite the article.

Reviewer 2 Report

The manuscript by Suzuki et al., used MD simulations to the study the mixed system of TIPS-P and atactic polystyrene (aPS) and understand the structure of the mixed thin film at the molecular level and the influence on the device properties. I agree for publishing this article in the present form.

Author Response

We appreciate the reviewers for taking their time in reviewing our manuscript.

Reviewer 3 Report

Comments on nanomaterials-2099675:

The current manuscript reports a computational study of some polymeric systems with atomistic simulations. The reported scientific data include correlation functions, density profiles, diffusion and many other routine properties describing the spatial arrangement of bulk liquids. Although the topic could be interesting to the nanomaterials audience, I identify a number of issues relating to presentation, scientific relevance and theoretical rigor. Significant modifications are expected in the revision stage, and the authors should really do some solid works to improve the quality of their work.

Im not sure why the abbreviation OTFT is defined. It is not cited at all. The same applies to many other cases such as AFM presented on the last paragraph of section 1.

Table 1 is useless. The authors are using well-pipelined program to generate the parameter files for systems under investigation. The force field parameters including those shown in Table 1 are automatically loaded from the pre-fitted force fields and are not fitted by the authors themselves. Thus, providing these data are scientifically useless and should be removed from the manuscript. Deleting these useless tables is also supported by the text presented later in the paper. The authors do not discuss about these force-field parameters in the results or discussions part, which indicates that these data are of little relevance to the scientific results presented in their paper.

Citations to the employed force-field parameters should be provided.

The definition of non-bonded interactions shown in Eq. (1) does not seem proper and requires significant clarifications and corrections. First, strictly speaking, the sigma parameter in LJ potential does not define the zero point but more specifically coordinates the minimum. Traditionally, the zero point of a inter-molecular potential is at r_ij -> infinity. Second, if following strictly the definition of this equation, the scaling factor f_ij should not exist at all. Specifically, the first sentence below Eq. (1) defines E_ab as the interaction energy between molecule a and molecule b. As a result, the ith atom from molecule a and the jth atom from molecule b cannot be bonded at all, not to mention the two or less bonds apart condition and the three bonds apart condition discussed later in this paragraph.

When computing various structural properties, the authors are using only a rather small subset of the whole sampling trajectory, which seems rather strange in molecular simulation. For example, for density calculations presented in section 3.3, the authors are using only 50 ns of their 1000 ns trajectory. Why? I do not see any scientifically solid motivation to perform such a contraction. The authors should report the relevant data extracted from the full-length trajectory and properly present the time-dependent behavior of the outcome for convergence check.

The core of the current manuscript is molecular dynamics simulation, which relies on statistical mechanics and approximates the ensemble averages with the time-averaged data. However, the authors do not properly define the statistical uncertainty for statistical observables reported in their work, which seems quite improper.  

Many equations are defined unnecessarily. For example, the OPLS function has a well-known form and is not newly introduced in this work by the authors. They are neither involved in discussions presented later and thus should be totally removed from the manuscript. As for the other properties such as MSD, they are also well-known observables and do not require explicit definition. Otherwise, it seems wired that the authors are not defining density profiles and many other ingredients involved in this work.

The simulation technique employed in the current study is rather simple and lacks a lot of details check. Simply using existing software to generate parameter files and them simulate the system could be quite easy. However, for a computationally orientated work we normally expect more detailed and insightful results. For example, the force-field parameters are employed without justification and validation, and the authors write with confidence that their simulation outcome is fully consistent with experiment. This is a rather dangerous situation that fortuitous error cancellations can happen. Even you get some properties similar to experiment, huge problems could exist for the other. On this aspect, the authors should at least check many properties of the system to validate the employed parameters. For instance, comparing the force-field energetics with the QM results would provide hints on the similarities and differences of the OPLS results and the high-level reference. Similarly, the authors could compare the atomic forces and many other properties as a check. The current results from simple program applications without careful validation are too crude to be predictable in the computational community.    

Finally, I would stress that the reviewers are volunteering their time to provide carefully summarized recommendations to improve the quality of the authors paper, and the authors should take these suggestions in the spirit in which they are intended. Do some solid works and improve your own manuscript. 

Author Response

Point 1: I’m not sure why the abbreviation OTFT is defined. It is not cited at all. The same applies to many other cases such as AFM presented on the last paragraph of section 1.

Response 1: We thank the reviewer for the helpful comment. We removed the abbreviations of OTFT, AFM, XRD, and PBC, which were cited only once or not cited.

Point 2: Table 1 is useless. The authors are using well-pipelined program to generate the parameter files for systems under investigation. The force field parameters including those shown in Table 1 are automatically loaded from the pre-fitted force fields and are not fitted by the authors themselves. Thus, providing these data are scientifically useless and should be removed from the manuscript. Deleting these useless tables is also supported by the text presented later in the paper. The authors do not discuss about these force-field parameters in the results or discussions part, which indicates that these data are of little relevance to the scientific results presented in their paper.

Response 2: We thank the reviewer for the helpful comment. We removed the tables of force-field parameters.

Point 3: Citations to the employed force-field parameters should be provided.

Response 3: The citations of [23], [24], and [25] contain the force-field parameters of aPS, TIPS-P, and toluene, respectively.

Point 4: The definition of non-bonded interactions shown in Eq. (1) does not seem proper and requires significant clarifications and corrections. First, strictly speaking, the sigma parameter in LJ potential does not define the zero point but more specifically coordinates the minimum. Traditionally, the zero point of a inter-molecular potential is at r_ij -> infinity. Second, if following strictly the definition of this equation, the scaling factor f_ij should not exist at all. Specifically, the first sentence below Eq. (1) defines E_ab as the interaction energy between molecule a and molecule b. As a result, the ith atom from molecule a and the jth atom from molecule b cannot be bonded at all, not to mention the ‘two or less bonds apart’ condition and the ‘three bonds apart’ condition discussed later in this paragraph.

Response 4: First, we checked Eq. 1 and the definition of sigma. It seems that the sigma is truly the distance where the interaction energy is zero with respect to the infinite distance as a zero reference. Substituting r = sigma gives E = 0. Instead, substituting r = 2^(1/6) * sigma gives the minimum energy. Second, we understand that the E_ab should not be defined as the intermolecular interaction energy because the E_ab also includes intramolecular interaction. Considering the comment below (Point 7), we finally removed the equations of the force field.

Point 5: When computing various structural properties, the authors are using only a rather small subset of the whole sampling trajectory, which seems rather strange in molecular simulation. For example, for density calculations presented in section 3.3, the authors are using only 50 ns of their 1000 ns trajectory. Why? I do not see any scientifically solid motivation to perform such a contraction. The authors should report the relevant data extracted from the full-length trajectory and properly present the time-dependent behavior of the outcome for convergence check.

Response 5: We thank the reviewer for the important suggestion. We added a plot of the time-dependence of MDS (Figure 3) for the full-length trajectory. The plot showed that the system with 50 w/w% toluene could be considered in the equilibrium state after 500 ns. The density profiles are also shown for t = 50, 500, and 1000 ns (Figure 5). The density profile at 500 ns was quite similar to that at 1000 ns.

Point 6: The core of the current manuscript is molecular dynamics simulation, which relies on statistical mechanics and approximates the ensemble averages with the time-averaged data. However, the authors do not properly define the statistical uncertainty for statistical observables reported in their work, which seems quite improper.  

Response 6: We thank the reviewer for the helpful comment. We hope that the new figures (Figure 3, 5, and 6) have improved the reliability of our results.

Point 7: Many equations are defined unnecessarily. For example, the OPLS function has a well-known form and is not newly introduced in this work by the authors. They are neither involved in discussions presented later and thus should be totally removed from the manuscript. As for the other properties such as MSD, they are also well-known observables and do not require explicit definition. Otherwise, it seems wired that the authors are not defining density profiles and many other ingredients involved in this work.

Response 7: We thank the reviewer for the helpful comment. We removed the force-field functions and the definitions of the MSD and the diffusion constant. The definitions of autocorrelation function and fitting function were included as inline equations.

Point 8: The simulation technique employed in the current study is rather simple and lacks a lot of details check. Simply using existing software to generate parameter files and them simulate the system could be quite easy. However, for a computationally orientated work we normally expect more detailed and insightful results. For example, the force-field parameters are employed without justification and validation, and the authors write with confidence that their simulation outcome is fully consistent with experiment. This is a rather dangerous situation that fortuitous error cancellations can happen. Even you get some properties similar to experiment, huge problems could exist for the other. On this aspect, the authors should at least check many properties of the system to validate the employed parameters. For instance, comparing the force-field energetics with the QM results would provide hints on the similarities and differences of the OPLS results and the high-level reference. Similarly, the authors could compare the atomic forces and many other properties as a check. The current results from simple program applications without careful validation are too crude to be predictable in the computational community.

Response 8: We thank the reviewer for the important comment. We understand that the validation of the classical force field simulation is quite important and that the simulation of the pristine TIPS-P crystal is not enough to fully validate the employed force field. Further validation of the simulation results such as comparing with QM results is an issue for us in the future.

Point 9: Finally, I would stress that the reviewers are volunteering their time to provide carefully summarized recommendations to improve the quality of the authors’ paper, and the authors should take these suggestions in the spirit in which they are intended. Do some solid works and improve your own manuscript.

Response 9: We deeply appreciate all the reviewers again for their time in reviewing our manuscript. We have made our best to improve our manuscript within the deadline. We hope the manuscript has been much improved owing to the reviewers’ precious suggestions.

Round 2

Reviewer 3 Report

Comments on nanomaterials-2099675.R1:

Its good to see that the quality of the manuscript has been improved with the efforts of the authors and its close to acceptance. Before that, the authors are recommended to make further efforts on the following points detailed below.

First, statistical uncertainties should be reported for statistical observables reported in this paper. Although I have pointed this out in my last review, the authors do not properly add this critical quantity. This time, I stress it again that all statistical observables extracted from a simulation trajectory, including those reported in Table 2, 3 and 4, the surface roughness in Figure 7 and the intensity in Figure 2, have statistical fluctuations and should have some estimates of their uncertainties.

Second, for pressure control the authors are using an algorithm with known flaws, the Berendsen barostat. The Berendsens strategy (either thermostat or barostat) has been recognized to produce deviations from the expectation for the targeted observable (temperature or pressure) and thus is currently not recommended. For the software that the authors are using, the Parrinello-Rahman regime and some alternatives are suggested in the manual. Therefore, I wonder why the authors are keeping using the outdated Berendsen barostat.

Round 3

Reviewer 3 Report

Comments on nanomaterials-2099675.R2:

The authors properly responded to my comments and the current version seems satisfactory. Before publication, I have only one small suggestion. The density comparison presented in the response letter (Berendsen vs Parrinello-Rahman) is very informative and validates the applicability of the weak coupling regime in practical non-aqueous simulations. This comparison deserves to be included in the main article, providing a crucial addition to literature supporting the usage of the Berendsens strategy. 

Author Response

Point 1: The authors properly responded to my comments and the current version seems satisfactory. Before publication, I have only one small suggestion. The density comparison presented in the response letter (Berendsen vs Parrinello-Rahman) is very informative and validates the applicability of the weak coupling regime in practical non-aqueous simulations. This comparison deserves to be included in the main article, providing a crucial addition to literature supporting the usage of the Berendsen’s strategy.

Response 1: We thank the reviewer again for the helpful suggestion. In the revised manuscript, we added a new figure (Figure 2) comparing the Berendsen and Parrinello-Rhaman, several sentences to explain the figure, and three references.